# Field Environmental Philosophy: A Biocultural Ethic Approach to Education and Ecotourism for Sustainability

Alejandra Tauro [1,2,*], Jaime Ojeda [1,3], Terrance Caviness [1], Kelli P. Moses [1], René Moreno-Terrazas [4], T. Wright [5], Danqiong Zhu [5], Alexandria K. Poole [6], Francisca Massardo [1] and Ricardo Rozzi [1,5,*]

1   Omora Ethnobotanical Park, Institute of Ecology and Biodiversity (UMAG-IEB), Universidad de Magallanes, Puerto Williams 6350000, Chile; jaimeojedav@gmail.com (J.O.); tccaviness333@gmail.com (T.C.); kelli.moses@gmail.com (K.P.M.); francisca.massardo@gmail.com (F.M.)
2   El Colegio de Puebla A.C., Puebla C.P. 72420, Mexico
3   School of Environmental Studies, University of Victoria, Victoria, BC V8W 2Y2, Canada
4   Departament of Humanities, Universidad Autónoma de Baja California Sur, La Paz C.P. 23080, Mexico; rtroyo@uabcs.mx
5   Subantarctic Biocultural Conservation Program and Department of Philosophy and Religion, University of North Texas, Denton, TX 76203-0920, USA; T.Wright@unt.edu (T.W.); Danqiong.Zhu@unt.edu (D.Z.)
6   Department of Philosophy, University of Twente, NB 7522 Enschede, The Netherlands; a.k.poole@utwente.nl
*   Correspondence: alejandratauro@gmail.com (A.T.); Ricardo.Rozzi@unt.edu (R.R.); Tel.: +1-940-565-2266 or +1-940-369-8211 (R.R.)

**Abstract:** To contribute to achieving local and global sustainability, we propose a novel educational methodology, called field environmental philosophy (FEP), which orients ecotourism practices to reconnect citizens and nature. FEP is based on the systemic approach of the biocultural ethic that values the vital links among the life *habits* of co-inhabitants (humans and other-than-humans) who share a common habitat. Based on this "3Hs" model (habitats, co-inhabitants, habits), FEP combines tourism with experiential education to reorient biocultural homogenization toward biocultural conservation. FEP's methodological approach seeks to integrate social, economic, and environmental dimensions of sustainability by generating new links between biological and cultural diversity at different spatial and social scales. Ecotourism has an underutilized potential to link sciences with education and conservation practices at different scales. By incorporating a philosophical foundation, FEP broadens both understanding and practices of environmental education and sustainable tourism. FEP has been developed at the Omora Ethnobotanical Park in the Cape Horn Biosphere Reserve, Chile, at the southern end of the Americas since 2000, where it has oriented transdisciplinary work for the creation of new protected areas and ecotourism practices. FEP enables an integration of biophysical, cultural, and institutional dimensions into the design of ecotourism activities that transform and broaden the perceptions of tourists, local guides, students, and other participants to better appreciate local biological and cultural diversity. FEP's methodology is starting to be adapted in other world regions, such as Germany, Japan, and Mexico, to integrate education and ecotourism for sustainability.

**Keywords:** biocultural conservation; biodiversity; Cape Horn Biosphere Reserve; Chile; ethics; metaphors; tourism

## 1. Introduction

During the twenty-first century, members of the globally interconnected society are increasingly losing direct everyday interactions with nature, leading to an "extinction of experience" [1–5]. Sustainable solutions are difficult to implement when local ecological knowledge and environmental sensitivity is threatened, as communities lose their understanding and engagement of what is needed to sustain ecological systems [6,7]. Further, an ecological understanding and ethical sense of responsibility for these losses is a necessary quality to maintain and nurture sustainability cultures. The biocultural discourse has

emerged seeking to expose the deep co-evolutionary interconnection between humans and their environments [8–10]. We argue that it is essential to identify key drivers that erode ecological knowledge and ethical valuing of biocultural diversity. To articulate the synchronous loss of biological, linguistic, and cultural diversity due to globally prevalent life habits and modes of production, we refer to this dynamic as "biocultural homogenization" [11]. Biocultural homogenization has multiple social-environmental drivers and consequences, such as the omission of local biodiversity, languages, and culture in school programs [12], losses of everyday knowledge about local birds and vegetation in cities [13], and native culinary habits and nutrition among indigenous communities in Chile [14]. Here, we articulate a novel methodology to address biocultural homogenization, and engage in developing ecological knowledge, environmental values, and sustainable lifestyle practices through a teaching and field practice, called "field environmental philosophy" (FEP). FEP orients formal and non-formal educational practices, including ecotourism, to overcoming the disconnection between citizen and nature.

FEP is a philosophical practice for epistemological and ethical reasons. We say epistemological because participants not only investigate biological and cultural diversity, but they also investigate the methods, languages, and worldviews through which scientific and other forms of ecological knowledge is forged. We say ethical because the aim is not only to research and learn about biological and cultural diversity but, foremost, to learn to respectfully co-inhabit within it. FEP is based on the systemic approach of the biocultural ethic that values the links between life habits of co-inhabitants (humans and other-than-humans) who share a common habitat [15]. Based on this "3Hs" model (habits, co-inhabitants, habitats), FEP combines tourism with experiential education to reorient biocultural homogenization toward biocultural diversity. FEP's methodological approach seeks to integrate social, economic, and environmental dimensions of sustainability by generating new links between biological and cultural diversity at different spatial and social scales. By incorporating a philosophical foundation, FEP broadens both the understanding and practices of sustainable tourism. This is especially relevant after the COVID-19 pandemic shut down the mass tourism industry in 2020.

The COVID-19 pandemic has highlighted the need to reorient certain tourism practices, and to design transformational experiences that restore the reconnection between humans and ecological systems [16,17]. It is indispensable to overcome an unsustainable tourism practiced globally for decades [18]. For example, by promoting local and close-to-home tourism, rediscovering the immediate surroundings where you live; by supporting innovative business models, especially from local economies, that respond to crises with cooperative actions and solidarity values; and also by promoting changes in the training of professionals towards responsible tourism, post-pandemic economies, and collaborative business models [18,19]. Tourism is, thus, seen as regenerative, and is complemented by the regenerative economy of local food systems and soil and planetary health, enhancing the positive impacts of a renewed industry [17]. It is a historic moment to shift human impacts towards a less intense, and more just distribution of the benefits of sustainable tourism. In a broader context, the World Tourism Organization maintains that sustainable tourism should comply with the UN Sustainable Development Goals (SDGs; www.unwto.org/tourism4sdgs, accessed on 10 February 2021). We assert that to effectively achieve a tourism that complies with the SDGs, it is necessary to broaden the understanding of ecotourism that defines biodiversity as a mere commodity, towards an understanding that integrates dimensions of justice, equity, and protection of the diversity of life forms, and highlighting the educational role of ecotourism [20,21]. However, we highlight the lack of (I) broader theoretical frameworks, and (II) methodologies to develop new forms of tourism that are responsive to the SDGs.

(I) Regarding theoretical frameworks, we offer a systemic and contextual definition for an ethical vision of ecotourism that fosters biocultural conservation. We start with the etymological and historical roots of the word "ecotourism," linking them to the conceptual framework of the 3Hs of the biocultural ethic. Ecotourism is a compound word deriving

from two terms. First, the Greek term *oikos*, which means home or habitat. Second, the French term tour, which means a journey that is associated with the habit of travelling. In essence, we propose an "oikos-tour," where:

> Ecotourism is an invitation to travel or tour to appreciate the life habits of human and other-than-human co-inhabitants in their oikos or local habitats, which brings well-being for those who visit and those who are visited ([22], p. 2).

This biocultural definition of ecotourism integrates multiple dimensions: (1) the biological and cultural diversity, and their interrelationships; (2) the uniqueness of the places combined with the tourism practices that enable its appreciation; (3) the well-being of the visitors and hosts; and (4) conservation of the destinations' "habitats". This approach could contribute to novel educational approaches to tourism that include environmental ethics with the goal of creating "virtuous tourists as agents of sustainability" [23]. The implementation of sustainable tourism requires a new educational approach [24–26] to generate changes in the attitudes of tourists. These changes include respect for the habitats and greater appreciation of the singularities of local biota and cultures by the tourists, as well as tour operators [23,24,27]. Environmental ethics can orient sustainable tourism. However, to implement this vision, we need to address the lack of methodologies that enable the integration of tourism and education for visitors, service providers, investors, and planners.

(II) Regarding methodological approaches that could contribute to an ecotourism education that integrates ecological sciences and environmental ethics, we present the FEP 4-step cycle. Until now, the integration of science, education, and ecotourism has been implemented in some Protected Areas and Biosphere Reserves across China [28], Mexico [29], Thailand [24], and some biological field stations, such as La Selva in Costa Rica or the Charles Darwin Station in Galapagos, have developed effective technical education programs for the members of the local community to be trained as research assistants [22,30–32]. This integration is essential to legitimize the development of tourism in protected areas, considering the perceptions that residents have about tourism and evaluating its positive and negative impacts [33].

FEP was developed at the southernmost point in the Americas, Omora Ethnobotanical Park, a Long-Term Socio-Ecological Research (LTSER) site within Chile's Cape Horn Biosphere Reserve (CHBR) [34]. The FEP approach integrates environmental ethics, arts, sciences, and education into (i) the co-production of new ecotourism themes and activities, and (ii) the training for tour guides from the local community to conduct special interest tours. We illustrate this approach with two case studies of the FEP models that could be adapted for educational and ecotourism practices in other regions of the world, particularly at LTSER sites and other research platforms, where the integration of science-education and ecotourism offers a great potential [34]. The two case studies are based on concepts developed in graduate courses and dissertations conducted at Omora Park, which have generated activities and other content, which are shared with tourists, school children, and other visitors.

## 2. Literature Review

Ecotourism offers an opportunity to integrate local social, environmental, and economic sustainability. The field of ecotourism, evaluated bibliometrically in the last 30 years, is shown to be associated with sustainable development [35,36]. Community-based ecotourism plays an especial key role in endogenous development, supporting the social and solidarity economy and the income of rural families in the long term [37–40]. However, the diversity of existing definitions of ecotourism characterize practices on a gradient from "genuine" sustainability and equity to "pseudo", "lite," and "greenwashing" [41]. Fennell identifies six key attributes to achieve a "genuine" ecotourism: (1) nature-based; (2) preservation; (3) education; (4) sustainability; (5) distribution of benefits; and (6) ethics/responsibility [41]. Among these attributes, education, despite being an essential requirement for ecotourism, has been less investigated than the other attributes of

"genuine" ecotourism [21,26,42–44]. Furthermore, a second gap about the study of ecotourism education derives from the tendency for research to have mostly focused on visitor learning [41,43]. However, teaching of guides also needs to be considered an essential component of ecotourism education, which includes empowerment of local community [21,27].

Regarding visitors' learning, a major research question concerns the effect of tourists' experiences in changing perceptions, values, knowledge, and attitudes towards pro-environmental behavior outside the visited place [23,25,44]. The predominantly cognitive and linear learning model, from guide to tourists, has been criticized because "there is a clear need for research and capacity building for guides as visitor educators in all forms of ecotourism" ([43], p. 29). Different types of ecotourism, for example, adventure or community-based, offer a series of activities and curricula that support different models of learning associated with different philosophical foundations [43]. For example, working with universities in Mexico and the United States, Bowan and Dallam [45] have developed a sustainable tourism education model using Fair-Trade principles and experiential learning philosophies that foster active engagement of students with farmers, fishers, hospitality providers, tourism outfitters, business owners, government officials, regional non-profits, and local citizens. Education via ecotourism experiences is not limited to only ecological or natural knowledge or the promotion of environmental behaviors, but it is also oriented towards knowledge and experimentation of the various ways of life and the biocultural diversity existing in the visited sites [21,27,46,47].

A greater emphasis on the type and quality of educational models used in the field of ecotourism could orient educational objectives toward more meaningful experiences that last beyond the sites visited. These experiences involve multidirectional interactions among tourists and guides, intertwined with their different life experiences, which can be enhanced by including a systemic and multifaceted vision of human beings in their journey. The learning obtained through the senses, through coexistence, through enjoyment and thought, all integrated in a series of methodologically guided activities can contribute to this end. At the same time, this approach could facilitate a rethinking of the aims of a linear ecotourism education towards a multidirectional one where tourists and visitors learn reciprocally from the exchanges in the travel encounters. Furthermore, it is imperative that these methodologies take in to account the complexity of cross-cultural interactions [47], to prevent commodifying ecotourism [20,26].

Regarding guides, they represent essential actors in the implementation of ecotourism [31,42], who interact with both visitors and the local communities that operate as service providers. Teaching guides involves formal and non-formal education. Professional guides are trained through formal education at universities or technical schools. In contrast, local guides are more often trained through non-formal or mixed types of education, typically offered by NGOs and in other workshop-style courses. Formal education emphasizes ecotourism training in university courses aimed at sustainability and ecotourism development [45,48,49]. Non-formal or mixed types of education for local guides and community operators focus on the development of diverse livelihoods and local empowerment [21,24,47]. To best incorporate education into the practice for a genuine ecotourism it needs to be considered that the types and models of education need to be adaptable according to different learners and their biocultural contexts.

For a more integral ecotourism education, trainers as well as curriculum and content developers should pay particular attention to the biophysical and cultural context where it takes place [22]. This emphasis on the biocultural context departs from standard formal education programs that often overlook the richness and diversity of forms of local knowledge embedded in biological and cultural diversity that is specific to locations in different regions of the world [50]. In Latin America, liberation philosophy emerged in the late 1960s as an intellectual response to the prevalence of decontextualized, universal, and Eurocentric traditions of thought [51]. This focus on local realities inspired Latin American educators, among them Paulo Freire. In 1970, after nearly two decades of literacy work in the Brazilian favelas and poor sectors of Brazil, he published his paradigm shifting

books, *Pedagogy of the Oppressed* [52] and *Education for the Praxis of Liberation* [53]. Freire introduced a new philosophy of education in which the learner rejects the notion of being a passive receptacle and instead embraces that learning is a dynamic process. In Freire's pedagogy of liberation teaching requires listening to the people. In the 1990s, this notion was expanded on by Leonardo Boff, a Brazilian liberation eco-theologian, who opens the demand to listen, not only to people of different cultures and social groups, but also to nature. Boff [54] affirms that not only the poor cry; but also the lands cry, the waters cry, nature cries. Dussel, Freire, and Boff provide philosophical foundations that have a great potential for contributing to a "genuine" ecotourism education that aims to: (i) be contextual, (ii) listen to culturally diverse humans perspectives and queues from biologically diverse organisms that co-inhabit places where ecotourism takes places, and (iii) foster an ethic that takes into account the actual biocultural diversity on which life depends, as well as social-environmental injustices that lead to oppression of less valued socio-cultural groups or appreciated biological species such as small invertebrates and non-vascular plants (e.g., mosses) [55]. Ecotourism education could benefit from liberation philosophies and methodological approaches that enhance the active participation of learners who train to become guides, and will later be in a position to enhance the active participation of visitors during their ecotourism experiences. This philosophical and pedagogical background has motivated the genesis of FEP's methodological approach that we present in this article.

## 3. Materials and Methods

### 3.1. Omora Park LTSER Site

In light of rapid global cultural, socioeconomic, and ecological transformations, LTSER sites can provide an institutional platform to conduct training and education, and to inform decision-making processes at multiple scales [56,57]. In this context, there was an urgent need to fill the geographical gap in the LTSER platform in the Global South, compelling Omora Park researchers to establish the Omora Park LTSER site in 2000 [34]. Omora Park is located in the world's southernmost city, Puerto Williams, (55.25° S 69.5° W, Figure 1). It was co-created by an interdisciplinary research team, with the participation by members of the local community, including Yaghan indigenous community, government authorities, navy personnel, and schoolteachers. The social component, "S," of this LTSER site is focused on the biocultural approach for integrating research, education, and conservation [34]. This integration was quickly applied as biocultural training for ecotourism guides, co-taught by philosophers, artists, and scientists. Following soon after, Omora Park research team initiated additional FEP programs for policy makers in 2001.

Over the last two decades, Omora Park has served primarily as a research and education center visited by an average of 1280 people annually. Most visitors are students or education-oriented groups of all ages; preschool to graduate level students (17%), local families (35%), and 36% are Chilean navy officials/personnel, decision makers, tourism operators/guides, and national and international researchers. The Omora Park has also offered guided visits for tourists, mainly from other parts of Chile, America, and Europe (12%), but Omora Park does not operate as a tourism service or provider. Thus, the vast majority (approximately 80%) of funding for operational and maintenance expenses are covered through research, education, and conservation projects and activities. Omora Park offers three interpretative trails in which FEP activities are conducted: (1) the "Southernmost Forests" trail, (2) "Underwater Inhabitants" trail, and (3) the "Miniature Forests" trail. The trails have a maximum capacity of 15, 7, and 10 visitors per group, respectively.

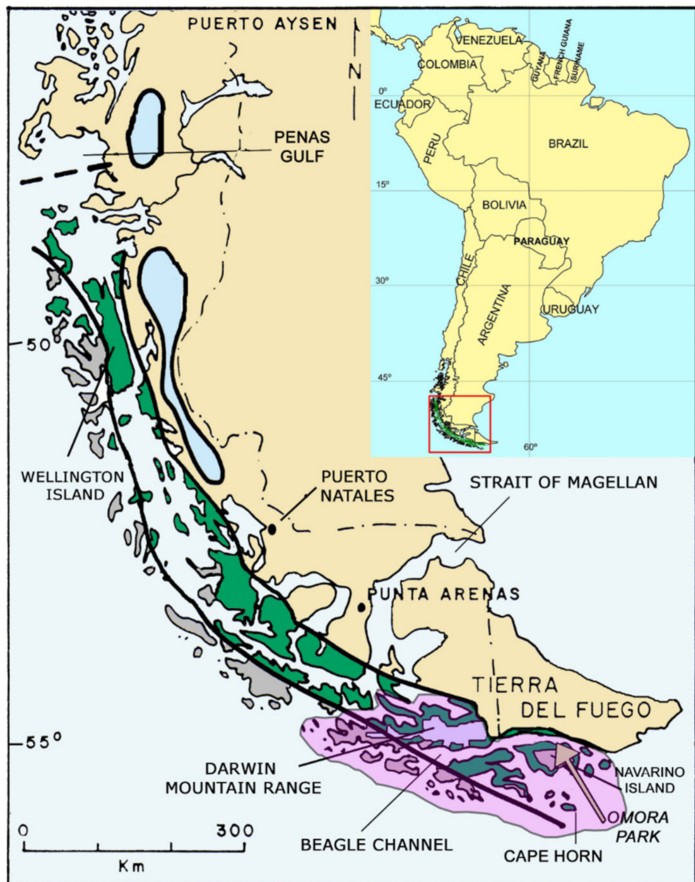

**Figure 1.** Map of the southern end of South America indicating the sub-Antarctic Magellanic ecoregion, showing the full extent of evergreen rainforests (green) and Magellanic moorland (gray) from Cape Horn to the Penas Gulf in Chile. Located on Navarino Island south of Tierra del Fuego, it is Omora Park that serves as the biocultural research, education, and conservation center of the Cape Horn Biosphere Reserve (indicated by light pink color). Figure modified from [34].

### 3.2. FEP Methodological Approach

The Omora Park research team created the Field Environmental Philosophy (FEP) methodological approach to catalyze transformational changes in curriculum and attitudes towards biocultural diversity in formal and informal education, and among tour guides, tourists, policymakers, and other citizens. In 2002, FEP was incorporated into the University of Magallanes (UMAG) curriculum, and in 2005 into the University of North Texas through the sub-Antarctic Biocultural Conservation Program [34] for graduate and undergraduate education. This inter-institutional platform has catalyzed the participation of students from more than 30 universities in Chile, USA, and other countries, such as Brazil, Mexico, Japan, Italy, and Germany. From 2000–2019, Omora Park researchers and graduate students have conducted an annual elective course at the local school, and have offered over 1000 training sessions, courses, and workshops. The class size and profiles of the participants vary in different groups and each year; however, on average there are 400 students from the local school, 80 children from two preschools, and 15 tour operators and guides that participate annually in courses and workshops.

### 3.3. FEP's Four-Step Cycle

The FEP approach is implemented through a multidirectional 4-step cycle that includes: (1) transdisciplinary research integrating ecology and philosophy; (2) poetic communication through the composition of metaphors and narratives; (3) design of ecologically- and ethically-oriented field experiences; (4) implementation of in situ conservation practices. Multiple forms of knowledge are necessary to solve social-environmental problems. For

this reason, findings and data generated through this process are co-produced with and for the community, including decision makers and government authorities, to yield practices and actions that address problems at distinct scales. This integration of theory and practice fosters an ethic of responsibility, community solidarity, and concern for the well-being of the ecosystems, including their human and other-than-human co-inhabitants [11].

Step 1—Transdisciplinary research. Graduate students conduct research on ecology and biological diversity, combined with philosophical analysis on epistemological and ethical dimensions. They also participate with community members and actors from different institutions, so that practical knowledge is incorporated from members of different disciplines, institutions, and sociocultural groups, who speak different languages and have different forms of ecological knowledge and practices. Comparative analyses are carried out to identify similarities and differences between different ways of knowing, valuing, and living with biocultural diversity.

Step 2—Poetic communication. Graduate students practice analogical thinking and composition of metaphors and narratives. Analogical reasoning is a cognitive underpinning of the ability to notice and draw similarities across contexts [58]. This is an essential ability for biocultural research and conservation practices [34]. Metaphors constitute cognitive-linguistic figures, which are part of the fundamental cognitive structure of humans in their everyday as in their scientific thought [59]. Hence, metaphors are not only an effective means for communicating with the public, but they are also effective for generating novel synthesis of cross-cultural and cross-disciplinary concepts. The practice of composing metaphors has two main goals: (i) to achieve conceptual syntheses of facts and values and practical syntheses of actions in biocultural conservation and education, including ecotourism; (ii) creation of stories, and mental images that enable intercultural dialogues, engagement with the general public, and sharing the results obtained in FEP step 1.

Step 3—Co-creation of field experiences guided with an ecological and an ethical orientation. Graduate students experience direct or "face-to-face encounters" with diverse co-inhabitants in their habitats. By integrating emotions and concepts derived from philosophical, scientific, and vernacular knowledge into field activities, graduate students experientially apprehend the concept of co-inhabitants through direct encounters. Graduate students design experiential environmental education and ecotourism field activities with members of all ages from the local community and visitors in such a way that all are enabled to connect with diverse co-inhabitants.

Step 4—In-situ conservation and/or restoration. To foster a sense of responsibility as citizens who are ecologically and ethically educated, and proactively participate in caring for biocultural diversity, FEP requires graduate students to be involved in biocultural conservation actions; for example, the design and implementation of interpretive stations along trails, or areas for the protection of the inhabitants, their habits, and habitats (native habitats, species, and ecological interactions).

### 3.4. The 3H's Biocultural Ethic and Ecotourism

In this article, we present concepts developed as part of the Long-Term Socio-Ecological Research (LTSER) program at Omora Park that have been transferred to tourism activities. A series of graduate courses and dissertations have used FEP's 4-step cycle (methodological framework) and the 3Hs model of the biocultural ethic (conceptual framework) to design new ecotourism activities that could orient relationships of reciprocity between those who visit (tourists) and those who are visited (the community of human and other-than-human co-inhabitant). For members of different social groups in the local tourist destination, this type of ecotourism fosters solidarity economy, and autonomy. For members of a globally interconnected society, it allows tourists to learn from different forms of knowledge and life habits through co-inhabitation experiences in unique places. With this "biocultural lens," ecotourism enables the discovery and appreciation of local life habits and singular co-inhabitants that often remain "invisible" to most citizens. During twenty years at the Cape Horn Biosphere Reserve, FEP based education and ecotourism practices have

contributed to overcoming the "invisibility" of a unique biological and cultural diversity. We illustrate these results through the case studies of "Ecotourism with a Hand-Lens" and "Open your Eyes to Dive" (Ojo, Bucea con Ojo, in Spanish). Both the methodological and conceptual frameworks illustrated in these cases were generated in association with graduate research and have been transferred into educational and ecotourism activities that have been conducted with groups of tourists, school children, and other visitors to Omora Park.

## 4. Results

### *4.1. Ecotourism with a Hand-Lens*

On 5 December 2001, Chilean authorities invited Ricardo Rozzi, Director of Omora Park, to give a keynote address at the ceremony opening the international passage between Puerto Navarino, Chile, and Ushuaia, Argentina, through the Beagle Channel. Aware that the opening of the passage would create opportunities for ecotourism activities, at the same time could threaten the fragile sub-Antarctic biodiversity, Rozzi, in turn, invited attendees to look at a moss-covered rock. Through actively looking at the rock with the help of a magnifying glass, attendees were able to appreciate a fascinating "microcosm." This singularity had remained overlooked, but visitors were able to begin appreciating it through guided observation. Additionally, this experience offered a powerful way to understand the fragility of mosses and the need for careful observation, valuation, and conservation of the biodiversity at the southernmost tip of the Americas. In this ceremony, Rozzi proposed ecotourism with a hand-lens as a novel sustainable tourism activity, which was subsequently elaborated, and implemented, by a team using FEP methodology at Omora Park (Supplementary Material S1). Omora Park researchers initiated a series of annual field courses on bryological sciences and conservation, including ecotourism with a hand-lens. Below, we present a concise synthesis of FEP's 4-step cycle to ecotourism with a hand-lens (Figure 2).

#### 4.1.1. Step 1: Transdisciplinary Research: Floristic Inventories, Naming and Conserving Bryophytes

Omora Park researchers have generated results in the biophysical, linguistic-cultural, and institutional-infrastructural dimensions of bryophyte conservation. Regarding biophysical dimensions, through floristic surveys at Omora Park researchers discovered that more than 5% of bryophyte (mosses, liverworts, and hornworts) and lichen species known worldwide are found in the Cape Horn Biosphere Reserve, in less than 0.01% of the earth's land surface [34]. Hence, Cape Horn represents a global "hotspot" for bryophyte diversity, a finding that provided a central argument for the creation of the Cape Horn Biosphere Reserve [60].

Regarding linguistic-cultural dimensions, Omora Park researchers found that most bryophytes and lichens found in Cape Horn, and other regions of the world lack common names. This presented a great challenge. However, some distinct species or groups of species of bryophytes and lichens are named in different indigenous languages. For example, epiphyte lichens of the genus Usnea have names in Yahgan (the language of Cape Horn's native indigenous people), Mapudungun (the language of the Mapuche indigenous people), Spanish (the language of Chilean colonizers), and English (the language of Anglican missionaries). Research on the etymological and historical meaning of scientific names in the original descriptions of this lichen, also revealed Chilean popular culture, as well as European, Asian, Middle Eastern, and other cultures (Figure 3, Supplementary Material S2).

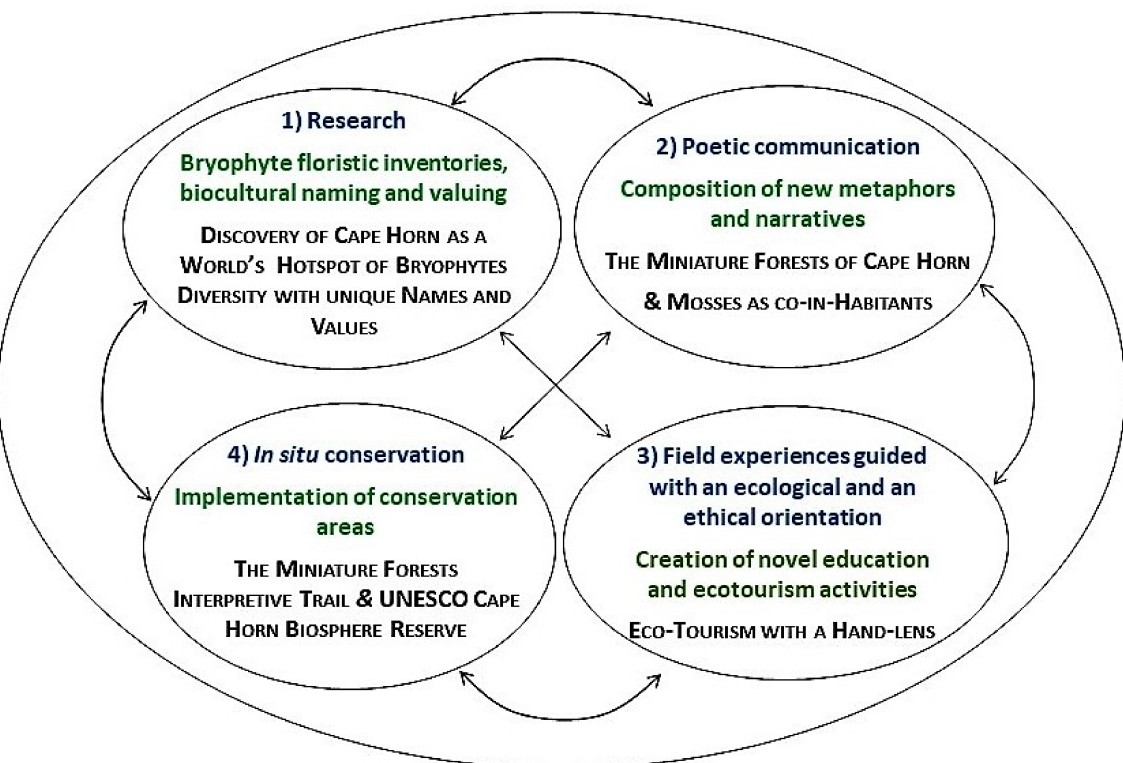

**Figure 2.** The Field Environmental Philosophy (FEP) 4-step cycle methodology adapted to the activity of "Eco-Tourism with a Hand-Lens". The methodological step is indicated in blue, the method used in green, and the results achieved in black. Arrows and lines indicate that interactions among the four steps are multidirectional.

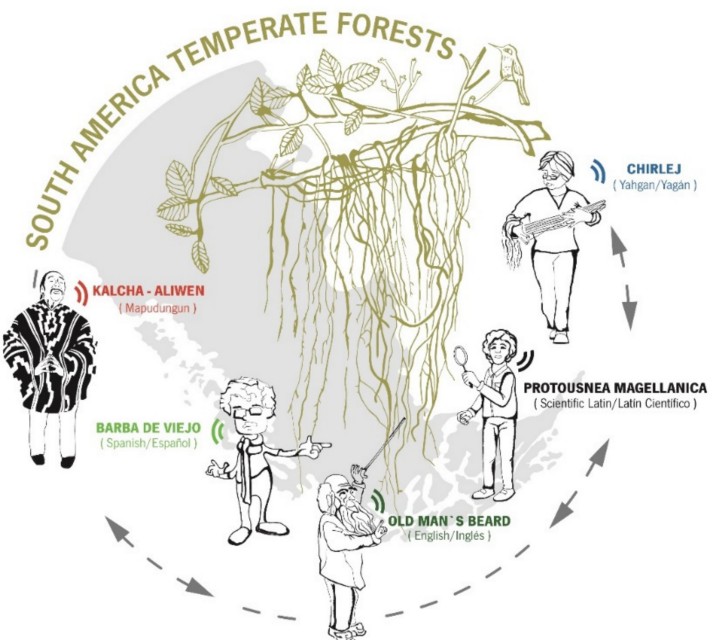

**Figure 3.** Representation of the diversity of languages that coexists in southwestern South America, isand involves a diversity worldviews, values, and ways of co-inhabiting with biological diversity [61]. For example, for a given type of lichen, in the indigenous Yahgan language, handcrafter Julia Gonzalez calls it "chirlej", and she maintains a unique ecological knowledge about its uses. Mapuche poet and Bird Man Lorenzo Aillapan names this type of lichens "kalcha-aliwen", where "kalcha" means hairs, and "aliwen" means tree. In English and Spanish, the names are "old-man beard" and "barba de viejo," respectively. These names have been used by naturalist Charles Darwin who extensively

traveled around southern South America and developed the mechanism of natural selection for his theory of evolution, and by Chilean biologist Humberto Maturana who proposed the alternative theory of natural drift for the evolution of life. The last three names express an analogy between the physical appearance of the lichen and the hairs of human. Regarding its scientific name, Dr. Susana Calvelo, an Argentine lichenologist renamed it recently as Protousnea magellanica [61]. "Proto-" means ancestral to Usnea, a genus of worldwide distribution whose name is derived from the Arabic "ushnah" or lichen; and "Magellanica" refers to its distribution throughout the Magallanes region.

Regarding policy dimensions, Omora Park researchers assessed how conservation priorities from the 1980s to the 2000s had severe taxonomic biases by relying almost exclusively on vascular plants and vertebrate fauna [62]. This finding exposed a colossal gap between conservation policies and the dominant component of flora at high-latitudes due to the omission of bryophytes from conservation policies. This omission was solved by successfully proposing a "change of lenses" to consider biome-specific indicator groups for designing effective conservation strategies.

### 4.1.2. Step 2: Composition of Metaphors and Communication: The Miniature Forests of Cape Horn

Bryophytes are largely unknown to decision makers and the public. This fact represented a challenge for communicating the discovery of high bryophyte diversity. To solve this puzzle, the creation of a metaphor was useful. The "Miniature Forests of Cape Horn" was created with elementary school children to indicate the diversity of mosses, liverworts, lichens, fungi, and the small invertebrates [12]. The analogy between these "miniature forests" and the larger forests that surround them facilitates an understanding about the ecological interactions occurring in the "miniature forests," understood as little ecosystems. FEP participants made comparisons between the roles played by mosses and lichens and those played by trees and shrubs in the larger forests, between the fauna found within the miniature forests and those in the macro forests. For example, springtails (Collembola) and mites may disperse moss propagules in ways that insects and birds pollinate or disperse seeds in a large forest. Mites, tardigrades, and other insects feed on mosses and the algae and fungi that grow in their miniature forests, similar to how herbivores, birds, and insects feed on the herbs, shrubs, fungi, and fruit of the larger forests [63,64].

With an ethical orientation, small plants and other organisms are observed as living beings that grow, reproduce, have the capacity to sense temperature, humidity, and in the case of invertebrates to experience pain [55]. To express this understanding, with school children, Omora Park researchers created the metaphor "people of the miniature forests" [61]. The diversity of organisms, small and large, is understood as co-inhabitants who have relationships of interdependence at multiple scales. Ecotourism with a hand-lens provides both "physical" and "conceptual" hand-lenses (such as metaphors) that help visitors to see the individuality and interdependence of mosses so that these little plants cease to be perceived as "homogeneous green blankets" and begin to be appreciated as diverse co-inhabitants.

### 4.1.3. Step 3: Guided Field Experiences with Ecological and Ethical Meaning: Ecotourism with a Hand Lens

Education and tourism programs at Omora Park emphasize direct encounters between human beings, mosses, lichens, birds, algae, rivers, and other components of the ecosystem. These experiences stimulate in situ perceptions and valuations of biological and cultural diversity, which evoke the biocultural definition of ecotourism as a journey (tour) that enables the appreciation of diverse co-inhabitants (including little plants and animals) with particular life habits in unique habitats or oikos, such as the miniature forests. With a magnifying glass in their hands, participants are invited to explore the diversity of organisms inhabiting these small habitats (Figure 2). Trained guides orient visitors through discussions that encourage them to perceive the miniature forests in ways analogous to perceiving large forests.

To differentiate and individualize bryophytes and their co-inhabitants, Omora Park guides invite visitors to engage in another key activity: drawing and naming plants. Participants are encouraged to fill a chart based on the "3Hs" of the biocultural ethic. This requires close observation of many different co-inhabitants of the miniature forests. The act of drawing induces visitors to pay closer attention to details that they often overlook. Indeed, the longer visitors sit and draw, the more details they will observe; consequently, each species becomes unique to them. After visitors have drawn mosses or lichens, Omora Park guides invite them to create a name for each of the species (Supplementary Material S3). Then, participants share their drawings and names with other members of the group and explain the details that caught their attention. Finally, participants compare their drawings and names with scientific ones, as well as with Mapudungun, Yahgan, Spanish, or English names (if they are known). Participants are invited to identify similarities and differences among their own names and those given by diverse cultures to experientially understand the notion of complementarity among different perspectives. This complementarity stimulates discussion about the ecological, economic, aesthetic, or ethical values of these little organisms. These discussions enhance participants' understanding and valuation of the miniature forests and prepare them to later appreciate the little co-inhabitants with whom they share their yards, parks, and regional habitats.

### 4.1.4. Step 4: Implementation of In-Situ Biocultural Conservation Areas: The Miniature Forests of Cape Horn Trail

The discovery of the high diversity of southern bryoflora and the development of "Ecotourism with a Hand-Lens" stimulated Omora Park's interdisciplinary research group to design the interpretive trail known as "The Miniature Forests of Cape Horn" [61]. Ecologists, philosophers, artists, and members of the Yahgan indigenous community built a network of approximately 2 km of trails with twenty interpretive stations. Sculptures with the shape of magnifying glasses orient visitors along a natural garden that protects bryophytes and their ecological interactions with insects, fungi, bacteria, water, and soil. This is the world's first botanical garden dedicated to bryophytes and integrating scientific criteria, biocultural conservation, education, ecotourism, and field environmental ethics [61].

The conceptual change of biodiversity "conservation lenses" also had implications at a larger scale by providing a central argument for UNESCO to create the Cape Horn Biosphere Reserve in 2005. This is the most extensive biosphere reserve in southern South America, and its creation was a novelty worldwide: this is the first protected area created based on the diversity of mosses, liverworts, and lichens. These small organisms had been previously omitted in international conservation, and a "change of lenses" to assess biodiversity in Cape Horn led to changes in conservation policies locally and internationally. Activities such as ecotourism with a hand-lens, and its associated conservation actions and policies, could be replicated in other regions of the world to bring attention to underappreciated groups of organisms.

### 4.2. Open Your Eyes to Dive ("Ojo, Bucea Con Ojo"): Habitats, Habits, and Co-Inhabitants under the Sea

In archipelagic regions, such as Cape Horn, flows of energy and matter among terrestrial, freshwater, and marine ecosystems are inextricably linked [60]. Hence, Omora Park researchers investigate biodiversity and conduct education and conservation activities in these three types of habitats. Human–biodiversity relationships, however, are not free of conflicts in these environments. In 2008, while working at the Fishery Institute of Chile in the Drake Passage and fjords of Cape Horn, marine biologist Jaime Ojeda witnessed unsustainable practices by industrial fisheries. This experience compelled him to shift his work toward conservation, and he applied to the Master Program in sub-Antarctic Sciences and Conservation, University of Magallanes. To develop his thesis proposal, Jaime used FEP's methodology at Omora Park and generated a novel diving activity to appreciate the rich biodiversity of kelp forests (Figure 4).

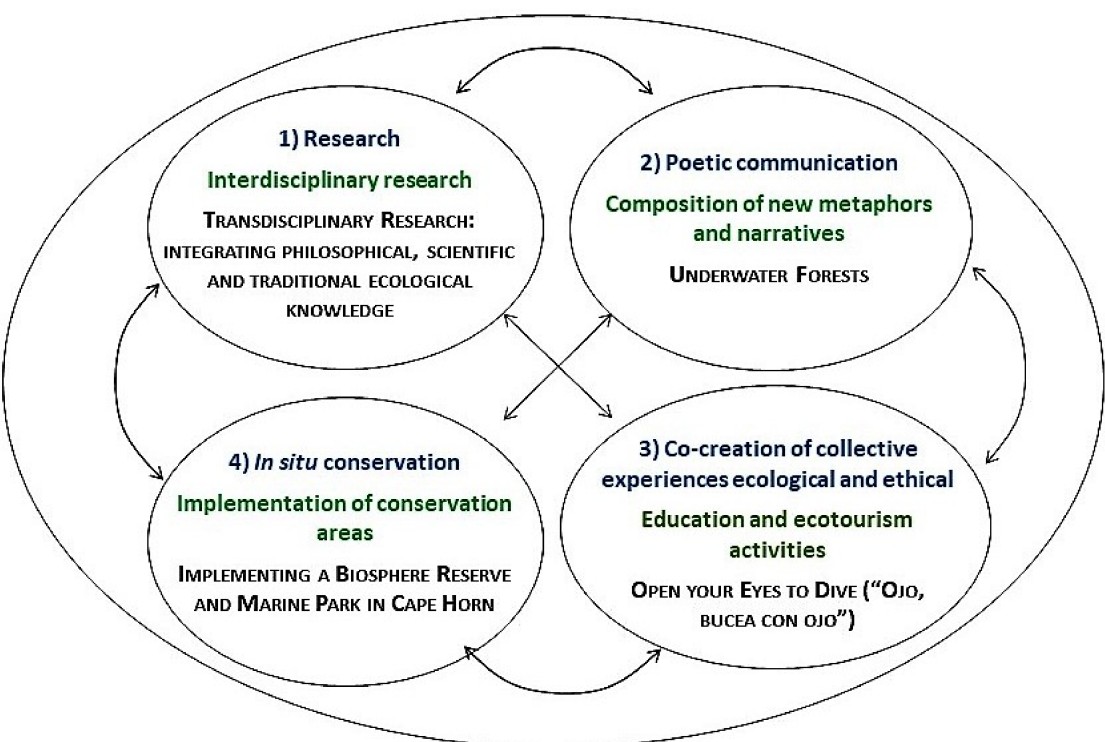

**Figure 4.** The Field Environmental Philosophy (FEP) 4-step cycle methodology to develop the activity of "Open your Eyes to Dive/Ojo, bucea con ojo". Each methodological step is indicated in blue, the method used in green, and the results achieved in black. Arrows and lines indicate that interactions among the four steps are multidirectional.

4.2.1. Step 1: Transdisciplinary Research: Integrating Philosophical, Scientific, and Traditional Ecological Knowledge

This research integrates three complementary "conceptual lenses:" the ecological, the philosophical, and the indigenous Yahgan worldview. Ecological research focused on biophysical dimensions, particularly the annual phenology of seaweeds and mollusks. In Cape Horn, coastal habitats are subject to oceanic climate conditions with a seasonality that is less pronounced than in subpolar latitudes in the Northern Hemisphere, which are subject to continental climate conditions. Hence, mollusks maintain relatively stable abundance throughout the year [65]. However, due to seasonal changes in the day-length, wet biomass of seaweeds is three times higher in summer than winter [66].

Regarding linguistic-cultural dimensions, Omora Park researchers identified that, by being abundant throughout the year and rich in minerals, mollusks have been a staple food for Yahgan indigenous people [65–67]. Mollusks have played a role analogous to bread for the Mediterranean peoples or corn in Mesoamerica. Yet, mollusks played another key role for the Yahgan culture: their shells provided the basic material to build the foundations for their huts or akar [67]. Shells on the hut's floors created good drainage conditions in rainy environments. As a protection against strong winds, Yahgan people also built walls with mollusk-shells placed around the bases of their huts. Today, thousands of shell middens are found along the coastal habitats of Cape Horn. Examined with the 3Hs model of the biocultural ethic, this rich archaeological heritage illustrates the intimate linkages between Yahgan life habits, the coastal habitats, and the mollusks as key biocultural co-inhabitants.

Regarding policy dimensions, Omora Park researchers have provided baseline information for determining the biocultural value of archaeological sites as well as for ethnographic studies that document the intimate connection of Yahgan people to their coastal habitats. The latter is important because this connection confers them priority rights for access to these coastal areas. The indigenous Yahgan community participated in the creation of the Cape Horn Biosphere Reserve [60], and is currently proposing the

establishment of coastal marine spaces for native people (Espacios Costeros Marinos Pueblos Originarios, ECMPO). Yahgan people have inhabited the archipelagoes of Cape Horn for over 6000 years [68], and documentation of their interspecific relationships of co-inhabitation with mollusks provides key information for conservation policies and projects that consider multiple ethical values, including the role played by mollusks in traditional Yahgan rituals [66,69] (Supplementary Material S4).

### 4.2.2. Step 2. Composition of Metaphors and Narratives: The Underwater Forests

Analogical thinking orients the poetic practice of creating metaphors through which FEP participants have represented submerged seaweeds as submerged forests. Analogous to terrestrial forests that present a marked seasonality in Cape Horn, seaweeds increase their biomass in synchrony with the sprouting of leaves on the trees of terrestrial forests during spring and summer. Based on this ecological analogy, the metaphor of "underwater forests of Cape Horn" facilitates an understanding about the ecological interactions occurring underwater, and enhances visualization through a mental image that is complementary to the more evident one of terrestrial forests [66].

The "underwater forests of Cape Horn" metaphor also facilitates understanding about the co-inhabitation relationships of the Yahgan people with marine living beings that have close kinship relationships with humans. Hence, for the Yahgans, mollusks have rights to occupy their habitats; for example, a limpet can be the owner of a rock. Consequently, Yahgan people believe that they ought to share intertidal habitats with mollusks and other marine organisms [66]. The "underwater forests" metaphor offers a "biocultural lens" to understand not only ecological relationships but also ethical interspecific relationships that have been oriented by ethical values and traditional ecological knowledge that originated in the remote region of Cape Horn, probably before the emergence of ancient Greek philosophy [55]. Today, this "biocultural lens" can inspire tourists to cultivate respectful forms of co-inhabitation with non-human organisms.

### 4.2.3. Step 3. Co-Creation of Ethical and Ecological Field Activities: "Open Your Eyes to Dive/Ojo, Bucea Con Ojo"

Educational programs at Omora Park emphasize direct encounters between human and other-than-human beings. To orient direct encounters, field activities are co-created with the participation of members of the local community. With the participation of Yahgan handcrafters, fishers, scientists, and philosophers, the activity "open your eyes to dive" (ojo, bucea con ojo in Spanish) was created. In Spanish "ojo" means "eye," and "pay close attention to something." This field activity brings together biological and cultural diversity, and their interrelations, and helps participants to experience ancestral and novel ways of seeing, feeling, and co-inhabiting with marine biodiversity.

"Open your eyes to dive" orients participants to having "face-to-face" encounters with invertebrates, seaweeds, and other organisms in the subtidal zone during high and low tides (Figure 3). This activity equips participants with two types of lenses: physical lenses (such as diving masks or filming cameras) and a conceptual lens, the "3Hs biocultural lens." Swimming among kelp forests, participants appreciate the habitats where invertebrates forge their life *habits* in interaction with other co-inhabitants. This biocultural lens orients participants to understand the concept of co-inhabitants through "face-to-face" encounters with the otherness of mollusks and other invertebrates in the submerged kelp forests of Cape Horn. By diving, participants experience a life habit that is essential for the culture of the Yahgan people and other fishers, as well as for invertebrates that co-inhabit in underwater forests of Cape Horn. By integrating emotions and concepts derived from indigenous, scientific, and philosophical knowledge into the observation of surprising and unexpected behaviors of invertebrates, students and other participants experientially apprehend the concept of co-inhabitants. The "open your eyes to dive" activity has triggered artistic inspiration and inspired novel environmental education programs, and today it offers a model for new community initiatives for underwater eco-tourism.

### 4.2.4. Step 4. Conservation in Situ: A New Marine Park in Cape Horn

The discovery of the close relationships between marine organisms and humans in the Cape Horn archipelagoes stimulated Omora Park researchers to lead the creation of marine protected areas. This initiative requires deep and systematic participation with various members of the communities and institutions involved in socio-environmental decision-making. Protecting marine ecosystems is essential to conserve experiences of co-inhabiting with algae, mollusks, and other beings (human and other-than-human beings) in the Cape Horn Biosphere Reserve. For this reason, in 2015, Omora Park researchers launched a complex participatory and policy processes to create the Diego Ramírez-Drake Passage Marine Park. Consistent with UNESCO's Man and Biosphere program that demands compatibility between social needs and conservation, the proposal maintained ancestral artisanal fishing practices and certified industrial fishing and ecotourism in way that is harmonious with the zoning scheme of the Cape Horn Biosphere Reserve. The process was not free of tensions. However, multi-stakeholder negotiations, including joint workshops with the Chilean Ministry of Economy and artisanal fishers as well as with industrial fishery, culminated in agreements about the design of the protected area [70]. In 2019, the Chilean government officially created the Diego Ramírez-Drake Passage Marine Park. This marine park protects critical nesting sites for albatrosses, penguins, and other endangered marine species, as well as intertidal and subtidal ecosystems with abundant macroalgae and invertebrates that reach their southernmost distribution [70]. In the context of climate change, with its abundant macroalgae and phytoplankton, this new marine park protects a significant carbon sink, a "blue lung" for the planet. In this way, FEP methodology and its participatory approach enabled the protection of Cape Horn's ecosystems, which represent one of the last wilderness areas remaining on the planet in the twenty-first century.

## 5. Discussion

In the remote region of Cape Horn, the vital links among human and other-than-human co-inhabitants have a long history, but today they are threatened by development models that neither consider nor value unique, diverse, and local biota and cultures. These development models are triggering processes of biocultural homogenization that involve simultaneous and interlocked losses of biological and cultural diversity at local, regional, and global scales [11]. Consequently, replacements of native biota and cultures by a globally uniform set of a few cosmopolitan, species, languages, and cultures are eliminating unique habitats, life-habits, and co-inhabitants [15]. To counterbalance processes of biocultural homogenization that are taking place in other regions of the world, we present the adaptation of the "3Hs" conceptual framework of the biocultural ethic to ecotourism practices that foster biocultural conservation and contribute to social and environmental sustainability.

The biocultural ethics methodology emphasizes the focus on cultivating an ethical sense of care that is embedded within the unique interrelation of biological, cultural, and linguistic diversity, as it is tied to place and its unique expressions of this biocultural diversity. We propose that this approach can facilitate developing local ecological knowledge and fostering sustainably focused value systems as a way of redressing the extinction of experience and deterioration of this connection to develop, maintain, and engage sustainable cultures. As such, we have been working with researchers and communities in multiple locations to engage in biocultural conservation in these unique communities. We next present some of these collaborations in the context of biocultural ethics and conservation.

### 5.1. Adapting FEP and the Biocultural Ethic's 3Hs Model to New Initiatives and Locations

Innovative life habits that are not only environmentally and economically sustainable, but at the same time ethically virtuous in their relationships with co-habitants, can be dynamic and have rapid responses in urban habitats. In Japan, for example, philosopher and FEP collaborator Mitsuyo Toyoda has led restoration programs of urban estuarine habitats on Sado Island. The restoration of estuarine habitat has, in turn, enabled the restoration of life habits associated with oyster fishing and education, which have catalyzed

the return of diverse co-inhabitants, including diverse forms of human cultures (e.g., fishers and other citizens of Sado Island) and biological species (e.g., oysters, wetland plants) [71]. To achieve estuarine restoration, it has been crucial to invite diverse people, such as fishers, farmers, government officials, schoolteachers, and even children, to participate and share their ideas [72].

With FEP collaborator Kurt Jax, we are initiating a project to examine reciprocal links between habitats and life habits in Germany's first protected area: the Drachenfels (Dragon's Rock) [11]. Located at the Rhine River, its establishment during the 1830s as a natural monument to protect the ruined Drachenfels castle illustrates that this conservation movement began not to protect "wild" landscapes but to protect biocultural landscapes in their homeland [73]. The place gained popularity, and a neo-Gothic castle was built lower on the hill in 1882. Because of the steep slope, a cog railway was constructed at the end of the nineteenth century to satisfy growing tourism demands. Today, tourism activities include wine tasting from grapes harvested by hand on the steep hillside. This case illustrates the intimate bonds between biological and cultural habitats and life habits, and how these bonds can be dynamically adapted to ecotourism to satisfy multiple needs.

In Mexico, philosopher and FEP collaborator René Moreno and his collaborators at the University of Baja California, La Paz, have developed a "Philosophical Hiking" university course [74]. In this elective course for the humanities and a degree in Alternative Tourism, participants explore the philosophical idea of Homo Viator, which understands journeying and traveling as substantive components of being properly human [75]. In a peripatetic way (in the tradition of Aristotle's pedagogical habit of teaching while walking), philosophical hiking is a communitarian "social practice" that articulates reason with emotion, nature with (collective and personal) culture. This FEP experience includes activities such as snorkeling in the Sea of Cortez followed by reflections oriented by the philosophical concept of Homo Viator. In this FEP activity, the metaphor of the "starry sea and sky of the Gulf of California" has been composed. Not only the sky has stars, the sea is also a landscape populated with them. Unlike the sky, in the sea we can reach them. Submerged in the sea we can perceive the microcosm in a drop of seawater or in the eye of an octopus, and then from the shore look up at the macrocosm to explore the celestial vault. All cultures have had symbolic representations of the sea and the sky, showing a unique biocultural richness. FEP methodology has stimulated a recovery and revaluation of Baja California myths that are little known by the participants, including the appreciation of how the stars of the sea and the sky have occupied a central place in the biocultural imagery of pre-Columbian and contemporary traditions [74].

The former examples illustrate how the 3H's model can be applied in different educational and ecotourism practices in contrasting regions of the world. Each location has its own biocultural identity that can be explored with field activities oriented by the 3Hs biocultural lens. These activities can be replicated in everyday locations such as urban parks and plazas, and rural neighborhoods. Recent studies have examined biocultural diversity in the urban ecosystems of Europe that are increasingly experiencing cultural heterogeneity in order to locate points of access for engaging with the natural and biocultural world [76]. An understanding of the local biocultural diversity of a locality is crucial to strengthening and building relationships between citizens and their local environment [2,6,7,77]. Additionally, recognition of urban biocultural diversity aids in overcoming the nature–culture dichotomy, and in cultivating traditional and novel relationships between local cultures and local biodiversity [76,78]. Education for sustainability needs to overcome a culture–nature dualism inherited from modernity, which still permeates globally interconnected society [79,80]. Those dissociated from nature inhabiting cities can find in ecotourism a possibility to reconnect with both biological and cultural diversity. FEP provides an accessible methodology to reconnect members of urban societies with local and regional ecosystems and cultural practices. Education should incorporate the wide recognition of biocultural diversity as a central element to achieve sustainability. Education for sustainability should

also balance theory with active co-creation and co-participation oriented by an ethics that allow us to rebuild relationships of co-inhabitation.

### 5.2. FEP: Integrating the Biocultural Ethic's 3Hs Model into Education and Ecotourism

FEP offers a novel methodology to link education and ecotourism for sustainability. Ecotourism, according to the International Ecotourism Society, should not only provide economic means for local people, but also share the socio-economic benefits among all involved parties [20]. Today, however, tourists often experience remote places through conventional cruises or all-inclusive resorts that offer staged images of nature with minimal environmental concern and nominal interaction with local communities [81]. Ecotourism offers an opportunity to visit, appreciate, and share the homes (or oikos) of diverse human and non-human co-inhabitants, their singular life habits, and habitats. It is problematic, however, that mass nature tourism eliminates access to these links, and generates a homogenized biocultural experience, socioecological degradation, and conspicuous distributive injustices, even in iconic places, such as Costa Rica, the Galapagos, and incipiently in Cape Horn [20,22]. To implement genuine ecotourism, it is necessary to overcome greenwashing marketing ambiguities, and pay close attention to local autonomy and biocultural diversity. FEP provides a biocultural lens through which to critically analyze tourism practices to better identify and evaluate extractive and mass tourism. Such tourism exoticizes biocultural diversity, exploiting it even without seeing and appreciating its uniqueness and values. It is essential to reorient non-sustainable and unjust practices labeled as ecotourism but which are disconnected and disrespectful of local communities and their habitats.

The biocultural ethic's 3Hs model has heuristic power for understanding the interrelationships among life habits, habitats, and the identity and well-being of the co-inhabitants of touristic destinations. Consequently, it more precisely generates principles of ecotourism that foster economic solidarity and demand social-environmental justice. In this way, the biocultural ethic highlights ecotourism's educational and socially transformative power. Local hosts transfer knowledge about their place to tourists by offering experiences in unique habitats, where diverse humans and other living beings cultivate unique life habits. These experiences reorient the preferences of visitors whose perceptions are broadened [27]. In addition to generating transformative experiences in the mindsets of visitors, it also generates economic revenues and ethical awareness.

FEP provides a biocultural lens through which to orient ethical ecotourism that contributes to social transformations favoring sustainability. We can find notable examples that are complementary to FEP's approach, such as the "geological walks" implemented in the Pyhä National Park in Finnish Lapland and the urban ecotourism "Naturewalks" in Barcelona, Spain. In the first example, visitors come closer to "mother Earth" without having to travel to far off exotic "wild" destinations [82]. Through local geotourism, they gain awareness of their proximity to planet Earth through notions such as rhythm, care, and vitality. This fosters domestic tourism and promotes geoethics [82]. In the second example, visitors pass through seldom-visited spaces in the city of Barcelona. By undertaking novel "Naturewalks," urban ecotourism helps to reconnect travelers with biological and cultural diversity by experiencing and learning about ethno-ecology, architecture, and history. Thus, visitors discover Barcelona in "a natural and subtle way" [83]. These examples show that ecotourism is both an external and an internal journey or *tour*, which allows reconnecting with other co-inhabitants and for visitors to reconnect with themselves.

Ecotourism is relevant not only for remote regions but also for urban and nearby rural locations. Biocultural diversity research has focused primarily on indigenous communities located in remote regions, but increasingly is giving attention to urban communities [9,78]. Urban societies host unique biological and cultural diversities involving novel and dynamic interactions [76]. Identifying biocultural singularities in urban and rural communities open opportunities for ecotourism, which require developing training practices as well as new concepts and activities for sustainable tourism [24]. FEP methodology can be used in a variety of urban, rural, and remote communities to orient the creation of novel ecotourism

narratives, educate broader audiences, and generate options for social, economic, and environmental sustainability across scales.

## 6. Concluding Remarks

We have argued that in order to foster sustainability it is necessary to engage environmental values, ecological knowledge, and lifestyle practices in addition to the physical drivers that cause biological diversity and ecosystem loss. An often-under-appreciated aspect of conservation efforts is the recognition of the importance of local ecological knowledge and fostering value systems that appreciate and engage nature. Through field environmental philosophy, that is, methods of engagement and knowing the environment in new and overlooked aspects with the use of science, the arts, and ethics, we can develop and foster biocultural ethics, or a sense of care that is focused on the unique interrelation of biological, cultural and linguistic diversity. This is a way of valuing that which is specific to each place and can be taken as a way of viewing the world that translates to a curiosity and engagement with the world wherever one finds oneself. Toward this aim, education plays a critical role, which can be extended to ecotourism. New educational approaches that include ethical reflection and a sense of responsibility for the conservation of biocultural diversity can orient novel forms of ecotourism. The FEP 4-step cycle offers a model for training in ecotourism, which brings together science, ethical engagement, and conservation of local biocultural diversity. Based on our FEP experiences, we identify five key lessons that add to a transdisciplinary methodological approach to foster forms of knowledge co-production and community engagement in ecotourism.

First, the FEP 4-step cycle in activities such as Ecotourism with a Hand-Lens and Open Your Eyes to Dive ("Ojo, bucea con Ojo") demonstrated the transformative power of experiences of distinguishing life habits, habitats, and co-inhabitants, and their interrelationships in unique habitats such as "miniature forests" or "underwater forests." FEP methodology stimulates an integration of education and ecotourism to experientially apprehend the concept of co-inhabitation on planet Earth.

Second, the creation of metaphors and narratives enhances the recognition of co-inhabitants whose existences often remain invisible. A clearer visualization of biological and cultural diversity catalyzes the cultivation of co-inhabitation bonds. In this way, FEP counteracts the denial, and consequent oppression, of the existences of myriads of living beings and cultural practices. FEP participants understand better the complexities of multi-dimensional and multi-scale processes involved in the pervasive problem of biocultural homogenization fostered by mass tourism.

Third, creative communication through analogical thinking aids our understanding of complex interrelationships among co-inhabitants who share a common habitat and have interdependent life-habits. Composition of metaphors associated with artistic, educational, and recreational actions form the basis for a sustainable ecotourism that catalyzes exchanges between hosts and visitors who value the diversity of humans and other organisms, such as mosses, lichens, and mollusks.

Fourth, FEP methodology can be applied in rural, urban, or natural parks around the globe. By combining the 3Hs model of the biocultural ethic with FEP's practical methodology, biophysical and cultural dimensions are integrated into the perceptions of students, tourists, local guides, and other participants who broaden their valuation of inconspicuous organisms and local forms of ecological knowledge. Our empirical results illustrate how this integration can be achieved through novel ecotourism activities.

Finally, FEP offers a valuable methodological approach to educate students, local guides, and other participants. To achieve FEP's approach, we highlight the importance of long-term transdisciplinary research platforms. Botanical gardens, biological stations, protected areas, and LTSER platforms can assist the monitoring of tourism practices and provide baseline information and concepts to orient regulatory frameworks to ensure a sustainable and ethical ecotourism. We have shown how place-based research initiatives can play a fundamental role in innovation and creation of novel ecotourism practices and

narratives that foster biocultural conservation and sustainability. At long-term transdisciplinary research and action platforms, the biocultural ethic's 3Hs model and FEP's practices offer a conceptual framework and a methodological approach to establish innovative education and ecotourism programs that contribute to social, economic, and environmental sustainability.

The World Tourism Organization is committed to responsible, sustainable and universally accessible tourism geared towards the achievement of the universal 2030 Agenda for Sustainable Development and the Sustainable Development Goals (SDGs). However, to achieve a tourism that complies with the SDGs, we state that it is necessary to develop broader theoretical frameworks and methodologies approaches. To address this need, we introduced the theoretical framework of the biocultural ethic and the methodological approach of Field Environmental Philosophy (FEP). This methodology can be adapted to range of environments throughout the rural-urban gradient and for all communities to aid in the engagement of biocultural diversity and foster a care for the deep interrelationship shared between human communities and their environment.

**Supplementary Materials:** The following are available online at https://www.mdpi.com/article/10.3390/su13084526/s1, Supplementary Materials S1 (SM1). Link to the documentary The Invisible Journey: Ecotourism with a Hand-Lens: https://www.youtube.com/watch?v=8Oxlhp3A-1s (accessed on 4 February 2021). This 19 min documentary, codirected by Jaime Sepúlveda and Ricardo Rozzi (2012), shows an innovative and sustainable tourist activity at the Omora Ethnobotanical Park: Ecotourism with Hand-Lens." It illustrates how the Field Environmental Philosophy methodology is applied on an ecotourism activity. This documentary is a journey with a narrative of natural history and ecology with an ecological and ethical guidance for the appreciation of the wonders of the "miniature forests" and its multiple ecological, aesthetic, economic, and ethical values. Exploring these forests generates a genuine ecotourism experience that links education, tourism, and sustainability sharing with human and other-than-human co-inhabitants in the Cape Horn Biosphere Reserve, Chile. Ecotourism with a Hand-Lens could be implemented in other regions of the world to appreciate components of biological and cultural diversity, and their interrelationships, which often remain invisible to globally interconnected society. Supplementary Materials S2 (SM2). Link to the brief documentary Old man's beard: the hairs of the sub-Antarctic forests: https://vimeo.com/494266072/ee40485eca (accessed on 4 February 2021). This 4 min documentary, codirected by Cristián Valle-Celedón, Alejandra Tauro and collaborators (2020), shows the interrelationships among diverse ways of naming old man's beard lichens and of co-inhabiting with them. This video is organized with the "3Hs" (habitats, habits, co-inhabitants) conceptual framework of the biocultural ethic to illustrate different types of knowledge. At the same time, it aims to contribute to biocultural conservation in southern South America, and also internationally to the Earth Stewardship Initiative of the Ecological Society of America (ESA) from the UNESCO Cape Horn Biosphere Reserve at the Southern End of the Americas. Supplementary Materials S3 (SM3). The Field Environmental Philosophy methodology includes composition of metaphors and drawing activities. In the case of the Ecotourism with a Hand-Lens, participants are invited to draw and create names for different types of bryophytes (mosses and liverworts) and lichens using the "3Hs" (habitats, habits, co-inhabitants) conceptual framework of the biocultural ethic. Each of the identified bryophytes and lichens is considered a co-inhabitant, for which participants identify a type of life habit and a micro-habitat (substrate; i.e., rock, forest, and tree bark) on which they grow as illustrated in Figure S1. Supplementary Materials S4 (SM4). Link to the brief documentary The Return to the Den (The Return to Ethics): https://vimeo.com/31905600 (accessed on 4 February 2021). This 25 min documentary, codirected by Jaime Sepúlveda and Ricardo Rozzi (2011), illustrates how the Field Environmental Philosophy methodology is applied on interdisciplinary and intercultural educational activities. The scene of Yahgan handcrafter Julia González and researcher Jaime Ojeda (minutes: 16:53–19:07) shows how in the Yahgan worldview, mollusks are perceived as co-inhabitants that punish selfishness, but honor respectful and collaborative habits.

**Author Contributions:** Conceptualization, A.T. and R.R.; methodology, A.T., F.M., J.O. and R.R.; formal analysis, A.T., T.C., J.O., D.Z., and R.R.; investigation, A.T., F.M., J.O. and R.R.; writing—original draft preparation, A.T.; T.C., K.P.M., D.Z., and R.R.; writing—review and editing, A.T., T.C., J.O., K.P.M., R.M.-T., T.W., D.Z., A.K.P. and R.R.; visualization, A.T., T.C., D.Z. and R.R.; supervision, R.R.; project administration, K.P.M.; funding acquisition, K.P.M. and R.R. All authors have read and agreed to the published version of the manuscript.

**Funding:** Funding was provided by Institute of Ecology and Biodiversity grant Basal Funding ANID-AFB170008, and IRES-National Science Foundation (NSF 1658651) awarded by the University of North Texas.

**Institutional Review Board Statement:** The study was conducted according to the guidelines of the Declaration of Helsinki, and approved by the Institutional Review Board (or Ethics Committee) following the guidelines of the Universidad de Magallanes (www.UMAG.cl, accessed on 4 February 2021) for graduate thesis and research, which include informed consents from human participants, and rules regarding collection protocols and treatment of animals.

**Informed Consent Statement:** Informed consent was obtained from all subjects involved in the study.

**Data Availability Statement:** The data presented in this study are available in supplementary material in this article, and further details are provided in the theses: Medina, Yanet. 2013. Bosques en Miniatura del Cabo de Hornos: el Turismo con Lupa como Herramienta para la Educación, Conservación y Turismo Científico en la Ecorregión Subantártica de Magallanes. Master in Sciences Program, Universidad de Magallanes, Chile. Ojeda, Jaime. 2013. Dinámica estacional de macroalgas y moluscos intermareales y su relación con el conocimiento tradicional ecológico yagán, en canales subantárticos del Cabo de Hornos: Una aproximación biocultural desde la filosofía ambiental de campo. Master in Sciences Program, Universidad de Magallanes, Chile.

**Acknowledgments:** We thank the valuable comments and editions suggested by Roy May on draft versions of this manuscript. Francisco Aguirre helps to improve our site study map. Vanessa Roman Abarca draws Figure 3. Paula Caballero y Jennifer Torres supported us with quantitative data of the Omora Park.

**Conflicts of Interest:** The authors declare no conflict of interest.

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
