# Peer review of "Field Environmental Philosophy: A Biocultural Ethic Approach to Education and Ecotourism for Sustainability"

_sustainability, doi:10.3390/su13084526_

Round 1
Reviewer 1 Report
The article is a worthy summary and analysis of ecotourism in practice, and one which is supported by an academic methodology of so-called Field Environmental Philosophy. If the following can be attended to, then the article would make a worthy contribution to the volume.
Overall, the article would benefit from closer and more sustained engagement with the sources cited. For example, the term "ethics" is clearly critical to the article, yet it is used without sufficient discussion of what it means in the context of the article.
It would improve also the readability of the article if the significance of the authors of some quotations were included.
The article could be improved by greater critical rigour – especially by interrogating notion of 'sustainability' and 'ecotourism' viz-a-viz the self-evident nature that the climate crisis has shown up how completely unsustainable all aspects of any industrial society are.
For instance, by definition, an economy dependent on external and remote resources (in this case, tourism) cannot therefore be ‘local.’ Nor can it therefore be 'sustainable', so, as admirable as the aspiration of 'ecotourism' is, the notion is based on greenwashing that the authors claim is solely the terrain of mainstream/mass tourism. Does ecotourism mean less transport to and from the destination to the home of the traveller(s)? Less ecologically destructive means of energy usage in the tourism activities themselves? So-called 'participants' fly to Chile (or any other site for that matter), travel around by petrol-powered vehicles in order to receive the experiences the authors detail. Any positive ‘local’ initiatives, as admirable as their intentions are, are unfortunately undermined by resource use, none of which can ever be truly sustainable.
The article requires a much clearer structure. For instance, it goes from " Step 4: Implementation of in-situ Biocultural Conservation Areas: The Miniature Forests 323 of Cape Horn Trail" to the next heading (or is it sub-heading? The formatting does not make it clear) of " 3.3. Open your eyes to dive (“Ojo, bucea con ojo”): habitats, habits, and co-inhabitants under the 345 sea" followed by the next sub/heading " Step 1: Transdisciplinary Research: integrating philosophical, scientific and traditional 362 ecological knowledge." Are the "Steps" sub-headings as per each 3.1, 3.2 etc? This needs to be made clear.
What, precisely, is the contribution that the article is making? Either to scholarship and/or to ecotourism practice. For instance, the last sentence of the abstract is unconvincing: " FEP’s methodology could be adapted in different world regions to effectively integrate education and ecotourism for 35 sustainability." Beyond "could", should it? If so, how, where, when and with whom?
Some claims require evidence to substantiate them, such as "Through close observation; drawing; naming; analogies and metaphors; and guided 307 discussion, participants begin to understand and identify the diversity of the miniature 308 forest and to appreciate and value this diversity." Same with " Participants reflects on the fact that 312 the oxygen they breathe is given to them by plants" - which participants? What is the sample size? Method? How did the researchers obtain this analysis? Arrive at this conclusion?
Similarly, "global society" is a vague and vacuous term - more Orwellian than something belonging in an academic paper. It should be replaced with a more accurate and less 'doublespeak' term.
There are numerous small mistakes in spelling, grammar and typos - for instance " Francisco Aguirre we help to improve our site study map." Some of these mistakes are much more serious - for instance, this sentence from the abstract makes no sense: " FEP combines tourism with experiential education to reorient biocultural homogenization toward biocutural homogenization."
Quotation marks are used where scare quotes should be used in-stead. For instance: " “think like a moss.” As no reference is made to Aldo Leopold's 'thinking like a mountain' , this statement can easily be interpreted as being a quote, whereas it is a paraphrase of Leopold's (unreferenced) land ethic.
The article is worthy of inclusion pending corrections identified above. These will require revisions and rewrites of some parts of the the article. It is not a revelatory analysis of ecotourism, yet it is a considered and methodical piece of research.
Author Response
REVISION NOTE
Manuscript: Field Environmental Philosophy: A Biocultural Ethic approach to education and ecotourism for sustainability
RESPONSES TO REVIEWER 1
We thank the reviewer for his/her thorough revision which helped us to improve the paper.
The article is a worthy summary and analysis of ecotourism in practice, and one which is supported by an academic methodology of so-called Field Environmental Philosophy. If the following can be attended to, then the article would make a worthy contribution to the volume.
Overall, the article would benefit from closer and more sustained engagement with the sources cited. For example, the term "ethics" is clearly critical to the article, yet it is used without sufficient discussion of what it means in the context of the article.
We added a concise explanation of our ethical and philosophical perspective in the Introduction (L65-69):
“FEP is a philosophical practice for epistemological and ethical reasons. We say epistemological because participants not only investigate biological and cultural diversity, but they also investigate the methods, languages, and worldviews through which scientific and other forms of ecological knowledge is forged. We say ethical because the aim is not only to research and learn about biological and cultural diversity but, foremost, to learn to respectfully co-inhabit within it.”
Ethics enters into the Field Environmental Philosophy (FEP) methodological approach both in its design and implementation. One example of this integration is in ‘face-to-face-encounters.’ We say that the biocultural ethic values the links among habits, habitats, and co-inhabitants; face-to-face encounters powerfully enact this valuation. Rather than focusing our published results on argumentative assertions related to specific philosophical questions considered in an abstract sense, we prefer to present the philosophical and ethical content of our educational ecotourism program.
It would improve also the readability of the article if the significance of the authors of some quotations were included.
We have removed some sentences with quotations to simplify the readability.
The article could be improved by greater critical rigour – especially by interrogating notion of 'sustainability' and 'ecotourism' viz-a-viz the self-evident nature that the climate crisis has shown up how completely unsustainable all aspects of any industrial society are.
We have added two paragraphs in the introduction (L79-106) that lay out critical views of sustainability and ecotourism in light of the changes triggered by the Covid-19 pandemic. The pandemic as a crisis has revealed structural problems at the global level and has strengthened the calls for changes, transformations and new practices that different populations have been demanding for decades and centuries in the world. We expanded the sustainability idea in regenerative economy and regenerative tourism in synergy with education and ethic.
For instance, by definition, an economy dependent on external and remote resources (in this case, tourism) cannot therefore be ‘local.’ Nor can it therefore be 'sustainable', so, as admirable as the aspiration of 'ecotourism' is, the notion is based on greenwashing that the authors claim is solely the terrain of mainstream/mass tourism. Does ecotourism mean less transport to and from the destination to the home of the traveller(s)? Less ecologically destructive means of energy usage in the tourism activities themselves? So-called 'participants' fly to Chile (or any other site for that matter), travel around by petrol-powered vehicles in order to receive the experiences the authors detail. Any positive ‘local’ initiatives, as admirable as their intentions are, are unfortunately undermined by resource use, none of which can ever be truly sustainable.
Among the paragraphs added in the introduction we present different conceptual views on ecotourism and sustainability. We are not unaware that there are unsustainable tourism practices or greenwashing (see reference to Fenell 2015), but rather we present the potential of ecotourism to support local communities in their development by diversifying their productive options. For this reason we support with literature on endogenous and regenerative tourism that assumes the generation of impacts of the activity but is committed to reducing the intensity, innovating practices and promoting an offer based on ethics and the responsibility of the traveler. It seeks to offset the effects of carbon emission in distributing the benefits under the principles of social and solidarity economy.
Additionally, we suggest that the fact that an increasing number of tourists are arriving from Punta Arenas, and other regions of Chile, Argentina, as well as more remote regions to Cape Horn requires that work is done together with local tour operators, guides, educators, and other members of local communities to orient tourism practices toward sustainable ecotourism. Consequently, we say: "FEP combines tourism with experiential education to reorient biocultural homogenization towards biocultural diversity." (L72-73). Our approach contributes with a practical method to better appreciate the uniqueness and biocultural diversity of places such as Cape Horn.
The article requires a much clearer structure. For instance, it goes from " Step 4: Implementation of in-situ Biocultural Conservation Areas: The Miniature Forests 323 of Cape Horn Trail" to the next heading (or is it sub-heading? The formatting does not make it clear) of " 3.3. Open your eyes to dive (“Ojo, bucea con ojo”): habitats, habits, and co-inhabitants under the 345 sea" followed by the next sub/heading " Step 1: Transdisciplinary Research: integrating philosophical, scientific and traditional 362 ecological knowledge." Are the "Steps" sub-headings as per each 3.1, 3.2 etc? This needs to be made clear.
We improved the structure adding sub-headings numbers in sub-section Results. The 3.3 was changed by 3.2 – listed above in errors to fix. 3. Results 3.1 Ecotourism with a Hand Lens 3.1.1 Step 1: Transdisciplinary, etc.
What, precisely, is the contribution that the article is making? Either to scholarship and/or to ecotourism practice. For instance, the last sentence of the abstract is unconvincing: " FEP’s methodology could be adapted in different world regions to effectively integrate education and ecotourism for 35 sustainability." Beyond "could", should it? If so, how, where, when and with whom?
We now clarify (L37-39): “FEP’s methodology is starting to be adapted in other world regions, such as Germany, Japan, Mexico, to integrate education and ecotourism for sustainability.”
Some claims require evidence to substantiate them, such as "Through close observation; drawing; naming; analogies and metaphors; and guided 307 discussion, participants begin to understand and identify the diversity of the miniature 308 forest and to appreciate and value this diversity." Same with " Participants reflects on the fact that 312 the oxygen they breathe is given to them by plants" - which participants? What is the sample size? Method? How did the researchers obtain this analysis? Arrive at this conclusion?
We adjusted some phrasing to clarify the methods and analysis in our case studies and the participants in it. For example, “The two case studies are based on concepts developed in graduate courses and dissertations conducted at Omora Park, which have generated contents and activities that are shared with tourists, school children and other visitors” (L145-147); or in each FEP’s step we change "participants" reference to “graduate student” who apply FEP methodology in tourist education context. So, when we say “participants” in each case studies, we refer to tourists, school children and other visitors at the Omora Park who participating in FEP activities.
Also we added one sentence about groups size: “The class size and profiles of the participants vary in different groups and each year; however, on average there are 400 students from the local school, 80 children from two preschools, and 15 tour operators and guides that participate annually in courses and workshops” (L190-193).
Similarly, "global society" is a vague and vacuous term - more Orwellian than something belonging in an academic paper. It should be replaced with a more accurate and less 'doublespeak' term.
We think that “global society” is an appropriate term for what is being referenced. The phrase is not uncommon. We think that likening to Orwellian doublespeak is a misrepresentation of what is actually written and its context and is more so a projection of the readers’ interpretation of the phrase. We re-phrasing it as “globally interconnected society”
There are numerous small mistakes in spelling, grammar and typos - for instance " Francisco Aguirre we help to improve our site study map." Some of these mistakes are much more serious - for instance, this sentence from the abstract makes no sense: " FEP combines tourism with experiential education to reorient biocultural homogenization toward biocutural homogenization."
Done. The manuscript was carefully reviewed by native English speakers.
Quotation marks are used where scare quotes should be used in-stead. For instance: " “think like a moss.” As no reference is made to Aldo Leopold's 'thinking like a mountain' , this statement can easily be interpreted as being a quote, whereas it is a paraphrase of Leopold's (unreferenced) land ethic.
We removed the text to simplify it.
The article is worthy of inclusion pending corrections identified above. These will require revisions and rewrites of some parts of the the article. It is not a revelatory analysis of ecotourism, yet it is a considered and methodical piece of research.
Thank you.
Reviewer 2 Report
The paper entitled "Field Environmental Philosophy: A Biocultural Ethic approach to education and ecotourism for sustainability" is a refreshing approaches to the topic of bio-cultural conservation park and eco-tourism.
The originality of the paper is to focus on bio-cultural horizontality - horizontality linked to the specificity the biotope concerned by the conservation-educational program: bryophyte and lichen species. Horizontality also of the local-to-local involvement of residents, students, autochthonous people and regional "tourists".
The paper is of good quality and its discussion encompassing comparisons with other experiences of the same kind in different part of the globe. Such a comparative perspective is very telling.
Some points can be improved.
First, there is a lack of quantitative data about this initiative.
How many people involve? Age? Social class? Ethnical background?
How many people visited the park? Is there a limitation of visitors? How many revenue is generated? How much of this private revenue is necessary for the park activities? How to measure the attractiveness of the park? How to equilibrate the need of visitors and their negative bio-ecological impact?
But finally the question: is it still about tourism?
Maybe eco-tourism should be clearer in its goal and declare "tourism" dead. The point here is the following: what is the meaning of local bio-eco-cultural sustainability if visitors come from far away places by planes, cars, etc. The fact that the park attracts foreign is it a sign of success or of internal contradiction? Is it not a way to outsource the CO2 emissions on the visitor? How ecological tourism can be really ecological without proscribing people to come from too far away places and restrict its opening to local people - i.e. without going against the very essence of modern capitalistic tourism.
Another point in the text is the usage of the term "philosophy": what is the philosophy involved in not really clear. Actually not clear at all. The term philosophy appears in the title and occurs in different places but with no conceptual definition at all. What kind of philosophy? P. 12 some details are provided about a philosophy of Hiking but it's more a practical theory of hiking than a philosophy and it's not related to the main example.
Then p. 10 there is this strange sentence: "The “underwater forests” metaphor offers a “biocultural lens” to understand not only ecological relationships but also ethical inter-specific relationships that have been oriented by ethical values and traditional ecological knowledge that originated in the remote region of Cape Horn, probably before the emergence of ancient Greek philosophy" How can you be sure about that? And is it important if it's before or after Heraclitus or Plato? What are the written or at least oral traces of this Cap Horn philosophy? Can you describe it and define its core principles and metaphysical values? Is it philosophy or a worldview? If a worldview: who is the seer? What are the anthropological evidences of it? There are references to "Analogical thinking" in other places of the paper but it remains extremely vague. Some development supported by Descola's categories will be helpful. Actually for Descola, analogical thinking is not related to Abogirinal knowledge described by the category of Animism regarding the Americas.
In other words, for a paper which harbors philosophy is in title, it lacks a consistent philosophical discourse that could provide the conceptual platform for the different experiences of bio-cultural conservation evoked. Even the notion of conservation should be better discussed: conservation? re-creation? appropriation?
Author Response
REVISION NOTE
Manuscript: Field Environmental Philosophy: A Biocultural Ethic approach to education and ecotourism for sustainability
RESPONSES TO REVIEWER 2
We thank the reviewer for his/her thorough revision which helped us to improve the paper.
The paper entitled "Field Environmental Philosophy: A Biocultural Ethic approach to education and ecotourism for sustainability" is a refreshing approaches to the topic of bio-cultural conservation park and eco-tourism.
The originality of the paper is to focus on bio-cultural horizontality - horizontality linked to the specificity the biotope concerned by the conservation-educational program: bryophyte and lichen species. Horizontality also of the local-to-local involvement of residents, students, autochthonous people and regional "tourists".
The paper is of good quality and its discussion encompassing comparisons with other experiences of the same kind in different part of the globe. Such a comparative perspective is very telling.
Some points can be improved.
First, there is a lack of quantitative data about this initiative.
How many people involve? Age? Social class? Ethnical background? How many people visited the park? Is there a limitation of visitors? How many revenue is generated? How much of this private revenue is necessary for the park activities? How to measure the attractiveness of the park?
We added one paragraph in Methods section to give key quantitative data about our activities (L162-172).
“Over the last two decades, Omora Park has served primarily as a research and education center visited by an average of 1,280 people annually. Most visitors are students or education-oriented groups of all ages; preschool to graduate level students (17%), local families (35%), and 36% are Chilean navy officials/personnel, decision makers, tourism operators/guides, and national and international researchers. The Omora Park has also offered guided visits for tourists, mainly from other parts of Chile, America and Europe (12%), but Omora Park does not operate as a tourism service or provider. Thus, the vast majority (approximately 80%) of funding for operational and maintenance expenses are covered through research, education, and conservation projects and activities. Omora Park offers three interpretative trails in which FEP activities are conducted: 1) the “Southermost Forests” trail, 2) “Underwater Inhabitants” trail, and 3) the “Miniature Forests” trail. The trails have a maximum capacity of 15, 7 and 10 visitors per group, respectively.”
How to equilibrate the need of visitors and their negative bio-ecological impact?
We added in the text one sentence in response to the question: “For example, by promoting local and close-to-home tourism, rediscovering the immediate surroundings where you live; by supporting innovative business models, especially from local economies, that respond to crises with cooperative actions and solidarity values; and also by promoting changes in the training of professionals towards responsible tourism, post-pandemic economies, and collaborative business models” (L82-86).
But finally the question: is it still about tourism?
Maybe eco-tourism should be clearer in its goal and declare "tourism" dead. The point here is the following: what is the meaning of local bio-eco-cultural sustainability if visitors come from far away places by planes, cars, etc. The fact that the park attracts foreign is it a sign of success or of internal contradiction? Is it not a way to outsource the CO2 emissions on the visitor? How ecological tourism can be really ecological without proscribing people to come from too far away places and restrict its opening to local people - i.e. without going against the very essence of modern capitalistic tourism.
Among the paragraphs added in the introduction we present different conceptual views on ecotourism and sustainability. We are not unaware that there are unsustainable tourism practices or greenwashing (see reference to Fenell 2015), but rather we present the potential of ecotourism to support local communities in their development by diversifying their productive options. For this reason we support with literature on endogenous and regenerative tourism that assumes the generation of impacts of the activity but is committed to reducing the intensity, innovating practices and promoting an offer based on ethics and the responsibility of the traveler. It seeks to offset the effects of carbon emission in distributing the benefits under the principles of social and solidarity economy.
Additionally, we suggest that the fact that an increasing number of tourists are arriving from Punta Arenas, and other regions of Chile, Argentina, as well as more remote regions to Cape Horn requires that work is done together with local tour operators, guides, educators, and other members of local communities to orient tourism practices toward sustainable ecotourism. Consequently, we say: "FEP combines tourism with experiential education to reorient biocultural homogenization towards biocultural diversity." (L72-73). Our approach contributes with a practical method to better appreciate the uniqueness and biocultural diversity of places such as Cape Horn.
Another point in the text is the usage of the term "philosophy": what is the philosophy involved in not really clear. Actually not clear at all. The term philosophy appears in the title and occurs in different places but with no conceptual definition at all.
We added a concise explanation of our ethical and philosophical perspective in the Introduction (L65-69):
“FEP is a philosophical practice for epistemological and ethical reasons. We say epistemological because participants not only investigate biological and cultural diversity, but they also investigate the methods, languages, and worldviews through which scientific and other forms of ecological knowledge is forged. We say ethical because the aim is not only to research and learn about biological and cultural diversity but, foremost, to learn to respectfully co-inhabit within it.”
Ethics enters into the Field Environmental Philosophy (FEP) methodological approach both in its design and implementation. One example of this integration is in ‘face-to-face-encounters.’ We say that the biocultural ethic values the links among habits, habitats, and co-inhabitants; face-to-face encounters powerfully enact this valuation. Rather than focusing our published results on argumentative assertions related to specific philosophical questions considered in an abstract sense, we prefer to present the philosophical and ethical content of our educational ecotourism program.
What kind of philosophy? P. 12 some details are provided about a philosophy of Hiking but it's more a practical theory of hiking than a philosophy and it's not related to the main example. Then p. 10 there is this strange sentence: "The “underwater forests” metaphor offers a “biocultural lens” to understand not only ecological relationships but also ethical inter-specific relationships that have been oriented by ethical values and traditional ecological knowledge that originated in the remote region of Cape Horn, probably before the emergence of ancient Greek philosophy" How can you be sure about that? And is it important if it's before or after Heraclitus or Plato? What are the written or at least oral traces of this Cap Horn philosophy? Can you describe it and define its core principles and metaphysical values? Is it philosophy or a worldview? If a worldview: who is the seer? What are the anthropological evidences of it? There are references to "Analogical thinking" in other places of the paper but it remains extremely vague. Some development supported by Descola's categories will be helpful. Actually for Descola, analogical thinking is not related to Abogirinal knowledge described by the category of Animism regarding the Americas. In other words, for a paper which harbors philosophy is in title, it lacks a consistent philosophical discourse that could provide the conceptual platform for the different experiences of bio-cultural conservation evoked. Even the notion of conservation should be better discussed: conservation? re-creation? appropriation?
We clarified the way in which we use analogical reasoning by adding this paragraph: “Graduate students practice analogical thinking and composition of metaphors and narratives. Analogical reasoning is a cognitive underpinning of the ability to notice and draw similarities across contexts [48]. This is an essential ability for biocultural research and conservation practices [46]. Metaphors constitute cognitive-linguistic figures, which are part of the fundamental cognitive structure of humans in their everyday as in their scientific thought [49]. Hence, metaphors are not only an effective means for communicating with the public, but they are also effective for generating novel synthesis of cross-cultural and cross-disciplinary concepts. The practice of composing metaphors has two main goals: (i) to achieve conceptual syntheses of facts and values and practical syntheses of actions in biocultural conservation and education, including ecotourism; (ii) creation of stories, and mental images that enable intercultural dialogues, engagement with the general public, and sharing the results obtained in FEP step 1” (L212-222).
In the Discussion section we explained the concept of Homo viator that supports the practice of philosophical hiking: “in Alternative Tourism, participants explore the philosophical idea of Homo viator, which understands journeying and traveling as substantive components of being properly human [69]. In a peripatetic way (in the tradition of Aristotle's pedagogical habit of teaching while walking), philosophical hiking is a communitarian "social practice" that articulates reason with emotion, nature with (collective and personal) culture” (L545-549).
Reviewer 3 Report
Comments regarding “Field Environmental Philosophy”
This paper self-identifies as a philosophy paper and the authors claim that its methodology was developed by a team that was one-third philosophers, artists and scientists, but there doesn’t seem to be any references to works related to the philosophy of biodiversity. It might be helpful to understand how the approach developed here builds upon or resolves prior philosophical approaches. There’s a running assumption here that this is the first of its kind, so there is nothing like it, but in fact I know of several philosophy papers whose claims and evidence would bolster their claims, and this connecting it to philosophical, just as they aim to reconnect people to their environments.
That said, it is a very enjoyable, well-written article. I see it as a strategy paper for reclaiming lands whose intense degradation risks prompting eco-refugees.
I found these few problems.
Line 24 should be either “biocultural diversity” or “biocultural convention,” not biodiversity homogenization.
Line 42 needs a period after [1,2]
Line 50 needs a “to overcoming” before “to overcome”
Lines 86 Please change: “On the one hand, there’s a lack of...”
Line 88 Please change: On the other hand, a pervasive consumerist tourism focuses on all-inclusive...”
Lines 392 and 411 interspecific doesn’t need a hyphen.
Line 545 “often” should follow “tourists”
Author Response
REVISION NOTE
Manuscript: Field Environmental Philosophy: A Biocultural Ethic approach to education and ecotourism for sustainability
RESPONSES TO REVIEWER 3
We thank the reviewer for his/her thorough revision which helped us to improve the paper.
This paper self-identifies as a philosophy paper and the authors claim that its methodology was developed by a team that was one-third philosophers, artists and scientists, but there doesn’t seem to be any references to works related to the philosophy of biodiversity. It might be helpful to understand how the approach developed here builds upon or resolves prior philosophical approaches. There’s a running assumption here that this is the first of its kind, so there is nothing like it, but in fact I know of several philosophy papers whose claims and evidence would bolster their claims, and this connecting it to philosophical, just as they aim to reconnect people to their environments.
We are aware of multiple philosophical papers that present and discuss concepts and practices to reconnect people to their environments. However, we reference only a few (Breakey and Breakey 2014) in our introduction in order to focus on the interface with ecotourism definitions. We present the biocultural ethic framework that support FEP methodological frameworks as a novel tool to reorient ecotourism practices towards bicultural conservation. The complementarity of the biocultural ethic with Leopoldian land ethic, Callicott’s environrmental ethic, Clare Palmer’s animal ethics, and other authors is extensively discussed in Ricardo Rozzi’s articles (2012, 2013, 2019).
We added a concise explanation of our ethical and philosophical perspective in the Introduction (L65-69):
“FEP is a philosophical practice for epistemological and ethical reasons. We say epistemological because participants not only investigate biological and cultural diversity, but they also investigate the methods, languages, and worldviews through which scientific and other forms of ecological knowledge is forged. We say ethical because the aim is not only to research and learn about biological and cultural diversity but, foremost, to learn to respectfully co-inhabit within it.”
Ethics enters into the Field Environmental Philosophy (FEP) methodological approach both in its design and implementation. One example of this integration is in ‘face-to-face-encounters.’ We say that the biocultural ethic values the links among habits, habitats, and co-inhabitants; face-to-face encounters powerfully enact this valuation. Rather than focusing our published results on argumentative assertions related to specific philosophical questions considered in an abstract sense, we prefer to present the philosophical and ethical content of our educational ecotourism program.
Finally, regarding the practice of philosophical hiking, we added in the Discussion section a concise explanation about the concept of Homo viator that supports this practice: “in Alternative Tourism, participants explore the philosophical idea of Homo viator, which understands journeying and traveling as substantive components of being properly human [69]. In a peripatetic way (in the tradition of Aristotle's pedagogical habit of teaching while walking), philosophical hiking is a communitarian "social practice" that articulates reason with emotion, nature with (collective and personal) culture” (L545-549).
That said, it is a very enjoyable, well-written article. I see it as a strategy paper for reclaiming lands whose intense degradation risks prompting eco-refugees.
Thank you, it is an interesting view.
I found these few problems.
Line 24 should be either “biocultural diversity” or “biocultural convention,” not biodiversity homogenization.
Done.
Line 42 needs a period after [1,2]
Done.
Line 50 needs a “to overcoming” before “to overcome”
Done.
Lines 86 Please change: “On the one hand, there’s a lack of...”
Done.
Line 88 Please change: On the other hand, a pervasive consumerist tourism focuses on all-inclusive...”
Done.
Lines 392 and 411 interspecific doesn’t need a hyphen.
Done.
Line 545 “often” should follow “tourists”
Done.
Reviewer 4 Report
In the manuscript the authors presented an important point, although the manuscript has some drawbacks.
Main remarks:
- Introduction - in the introduction, please write more about sustainability, tourism/ecotourism development. The manuscript was sent to Sustainability. Suggested publications:
- Roman, M .; Roman, M .; NiedzióÅ‚ka, A. Spatial Diversity of Tourism in the Countries of the European Union. Sustainability 2020, 12, 2713. https://doi.org/10.3390/su12072713
- Gu, X .; Hunt, C.A .; Lengieza, M.L .; Niu, L .; Wu, H .; Wang, Y .; Jia, X. Evaluating Residents ’Perceptions of Nature-Based Tourism with a Factor-Cluster Approach. Sustainability 2021, 13, 199. https://doi.org/10.3390/su13010199
2. Literature review - after the introduction there should be "literature review".
3. Conclusions - in conclusions, please also answer the following questions:
• what are the research gaps?
• what is new to this manuscript?
4. References - 26 out of 57 articles in the reference list are by R. Rozzi. This should not be done. They are self-citations ! ! !
Author Response
REVISION NOTE
Manuscript: Field Environmental Philosophy: A Biocultural Ethic approach to education and ecotourism for sustainability
RESPONSES TO REVIEWER 4
We thank the reviewer for his/her thorough revision which helped us to improve the paper.
In the manuscript the authors presented an important point, although the manuscript has some drawbacks.
Main remarks:
Introduction - in the introduction, please write more about sustainability, tourism/ecotourism development. The manuscript was sent to Sustainability. Suggested publications:
Roman, M .; Roman, M .; NiedzióÅ‚ka, A. Spatial Diversity of Tourism in the Countries of the European Union. Sustainability 2020, 12, 2713. https://doi.org/10.3390/su12072713
Gu, X .; Hunt, C.A .; Lengieza, M.L .; Niu, L .; Wu, H .; Wang, Y .; Jia, X. Evaluating Residents ’Perceptions of Nature-Based Tourism with a Factor-Cluster Approach. Sustainability 2021, 13, 199. https://doi.org/10.3390/su13010199
Thanks for the papers suggested, we added Gu et al 2021 (L135-137) plus others papers to explain the sustainability and ecotourism development. We developed these ideas in new Introduction paragraphs (L 79-106).
- Literature review - after the introduction there should be "literature review".
We included new paragraphs and we re-wrote some original paragraphs in our introduction to show this issue and don’t lose our original structure (L 91-138).
- Conclusions - in conclusions, please also answer the following questions:
- what are the research gaps?
- what is new to this manuscript?
We added in the Introduction section explicitly the gaps (L105-106) and we develop it (L107-137). We addressed it in the Conclusion section this gaps, we included new paragraphs in L629-638, L677-685.
- References - 26 out of 57 articles in the reference list are by R. Rozzi. This should not be done. They are self-citations ! ! !
We have extensively documented much of our philosophical reflection, practice, conversation, study, research, and more at Cape Horn, Texas, Germany, Mexico and elsewhere and we would be delighted if readers were interested in following our established history of relevant published philosophical contributions. Now, we have reduced to 20 own references and we have included many resources from a range of fields, including ethnoecology, conservation biology and environmental education, we have 78 total references.
Round 2
Reviewer 4 Report
- Rozzi's 20 publications are too many in a manuscript. These practices are inappropriate. In my opinion, this is the strength of placing your own scientific publications. It looks bad if the authors insert so many of their scientific publications. Whoever wants to quote such publications will quote it anyway.
- after the introduction, a part of "literature review" is missing.
Author Response
RESPONSES TO REVIEWER 4
We thank the reviewer for his/her thorough revision which helped us to improve the paper.
Comments and Suggestions for Authors (Reviewer 4)
Rozzi's 20 publications are too many in a manuscript. These practices are inappropriate. In my opinion, this is the strength of placing your own scientific publications. It looks bad if the authors insert so many of their scientific publications. Whoever wants to quote such publications will quote it anyway.
We reduced the list of references to include only 11 references by this coauthor. We consider that these references provide essential background methodological information and research that has not been conducted elsewhere. The methodological approach and conceptual framework presented in this article is based on and expands on the work that has been led by Ricardo Rozzi and his team. Furthermore, we use our references to introduce the implementation of two study cases developed at Omora Park.
after the introduction, a part of "literature review" is missing.
We have incorporated a new Literature Review section in the manuscript in which we provide background on education for ecotourism. In this review, we clarified the need for consideration of education processes for both ecotourism guides and visitors. We also specifically discussed liberation philosophy as a Latin American school of thought that provides philosophical and pedagogical foundations for our Field Environmental Philosophy Methodological approach.
Round 3
Reviewer 4 Report
- not all of my comments were included in the manuscript
- I never support writing articles and placing in them my 20 or more scientific publications,